# Coupling and uncoupling of midline morphogenesis and cell flow in amniote gastrulation

Rieko Asai[1], Vivek N Prakash[2], Shubham Sinha[2], Manu Prakash[3], Takashi Mikawa[1]*

[1]Cardiovascular Research Institute, University of California, San Francisco, San Francisco, United States; [2]Department of Physics, University of Miami, Coral Gables, United States; [3]Department of Bioengineering, Stanford University, Stanford, United States

*For correspondence:
takashi.mikawa@ucsf.edu

Competing interest: The authors declare that no competing interests exist.

**Abstract** Large-scale cell flow characterizes gastrulation in animal development. In amniote gastrulation, particularly in avian gastrula, a bilateral vortex-like counter-rotating cell flow, called 'polonaise movements', appears along the midline. Here, through experimental manipulations, we addressed relationships between the polonaise movements and morphogenesis of the primitive streak, the earliest midline structure in amniotes. Suppression of the Wnt/planar cell polarity (PCP) signaling pathway maintains the polonaise movements along a deformed primitive streak. Mitotic arrest leads to diminished extension and development of the primitive streak and maintains the early phase of the polonaise movements. Ectopically induced Vg1, an axis-inducing morphogen, generates the polonaise movements, aligned to the induced midline, but disturbs the stereotypical cell flow pattern at the authentic midline. Despite the altered cell flow, induction and extension of the primitive streak are preserved along both authentic and induced midlines. Finally, we show that ectopic axis-inducing morphogen, Vg1, is capable of initiating the polonaise movements without concomitant PS extension under mitotic arrest conditions. These results are consistent with a model wherein primitive streak morphogenesis is required for the maintenance of the polonaise movements, but the polonaise movements are not necessarily responsible for primitive streak morphogenesis. Our data describe a previously undefined relationship between the large-scale cell flow and midline morphogenesis in gastrulation.

## eLife assessment

Large scale cell movements occur during gastrulation in vertebrate embryos but their role in this major morphogenetic transition in formation of the body plan is poorly understood. Using the chick embryo model system, this study makes **important** advances using elegant methods to show that extension of the primitive streak during gastrulation, occurring through cell proliferation, polarisation and intercalation, and large-scale polonaise cell movements, can be uncoupled. Although the driving mechanism and precise role of these movements remains a mystery, the study provides **convincing** evidence for the uncoupling through independent approaches, the most creative of which are the effects shown after induction of a supernumerary primitive streak.

## Introduction

Large-scale cell flow during gastrulation is an evolutionarily conserved biological phenomenon in embryogenesis (*Gilbert and Barresi, 2017*; *Solnica-Krezel and Sepich, 2012*; *Leptin, 2005*). During this period, animals with bilateral symmetry, called bilaterians, initiate development of three germ

layers (i.e. the ectoderm, mesoderm and endoderm) and midline structures (e.g. the notochord and primitive streak) along the midline axis (*Solnica-Krezel, 2005*; *Keller and Davidson, 2004*; *Mikawa et al., 2004*). The large-scale cell flow has been linked to early embryonic morphogenesis, including axial structures, through rearrangement of cells and/or transportation of signaling molecules (*Solnica-Krezel and Sepich, 2012*; *Leptin, 2005*). Coupling of the large-scale cell flow and body axis morphogenesis has been extensively studied in invertebrate bilaterians, particularly insect models (*Keller, 2002*; *Gehrels et al., 2023*; *Bertet et al., 2004*). However, in vertebrates, our understanding of whether and how the large-scale cell flow and midline morphogenesis are coupled is limited.

Vertebrates are classified into two groups: non-amniotes (e.g. fish and amphibians) and amniotes (e.g. birds and mammals) (*Gilbert and Barresi, 2017*). In non-amniotes, particularly amphibians, the large-scale cell flow is coupled with formation of the notochord through Wnt/planar cell polarity (PCP) pathway-regulated convergent extension, which does not depend on mitosis (*Keller et al., 2000*; *Tada et al., 2002*). In amniotes, particularly in avian gastrula (i.e. embryonic disc), a bilateral vortex-like counter-rotating cell flow, termed 'polonaise movements', occurs within the epiblast along the midline axis, prior to and during primitive streak (PS) formation (*Wetzel, 1929*; *Gräper, 1929*; *Stern, 2004*). The PS is an evolutionarily unique embryonic structure in amniotes, which is not only an organizing center for gastrulation but also the earliest midline structure (*Mikawa et al., 2004*; *Schoenwolf, 2000*; *Raffaelli and Stern, 2020*; *Sheng et al., 2021*). During amniote gastrulation, widespread mitosis has been identified throughout the embryo, including the PS (*Sanders et al., 1993*), and previous studies showed that mitotic arrest leads to morphologically diminishing PS extension (*Cui et al., 2005*; *Saadaoui et al., 2020*). Further, the Wnt/PCP pathway also plays an important role for proper PS extension and patterning (*Voiculescu et al., 2007*). Since the chick embryo is readily accessible and closely represents the human gastrula (*O'Rahilly and Müller, 2010*; *Asai et al., 2021*), much of what is known about the cell flow and PS development has come from studies using this model system (*Stern, 2004*). The PS is first visible as a cell cluster at the posterior embryonic disc at the initial streak stage [HH2; staging (*Hamburger and Hamilton, 1992*)] formed through the influence of several axis inducing factors [e.g. Vg1 and Chordin (*Seleiro et al., 1996*; *Streit et al., 1998*)]; it then extends anteriorly along the midline axis (*Mikawa et al., 2004*; *Stern, 2004*; *Schoenwolf, 2000*). During PS development, displacement of cells and/or tissue tension potentially generates biomechanical forces, which lead to cell flows and/or PS morphogenesis (*Saadaoui et al., 2020*; *Voiculescu et al., 2007*; *Chuai et al., 2023*; *Rozbicki et al., 2015*). Recent studies discuss that cell intercalation and ingression underlying PS formation, leading to the polonaise movements through altering the neighbor tissue tension (*Voiculescu et al., 2007*; *Rozbicki et al., 2015*; *Voiculescu et al., 2014*), suggesting that the polonaise movements are likely a consequence of PS morphogenesis (*Raffaelli and Stern, 2020*; *Voiculescu, 2020*; *Wang and White, 2021*). Other studies demonstrate that the polonaise movements, generated by the myosin-cable-mediated tissue tension with cell intercalation and ingression, lead to PS morphogenesis (*Saadaoui et al., 2020*; *Chuai et al., 2023*; *Wang and White, 2021*). Thus, the relationship between the polonaise movements and PS morphogenesis remains controversial (*Wang and White, 2021*).

Here, we characterize previously undefined relationships between the polonaise movements and PS morphogenesis. Suppression of the Wnt/PCP pathway generates a wider and shorter PS while maintaining the polonaise movements. Mitotic arrest diminishes PS extension and maintains the early phase of the polonaise movements. Moreover, experimentally manipulating the large-scale cell flow by inducing a secondary midline axis indicates that the authentic PS forms despite altered cell flows. Ultimately, a single axis-inducing morphogen, Vg1, is capable of initiating the polonaise movements despite defective PS morphogenesis. These results suggest that PS morphogenesis is not responsible for the initiation and early phase of the polonaise movements. Further, the polonaise movements are not necessarily responsible for PS morphogenesis. This study suggests that amniote gastrulation may be evolutionarily distinct from non-amniote gastrulation in the mechanisms coupling large-scale cell flow and midline morphogenesis.

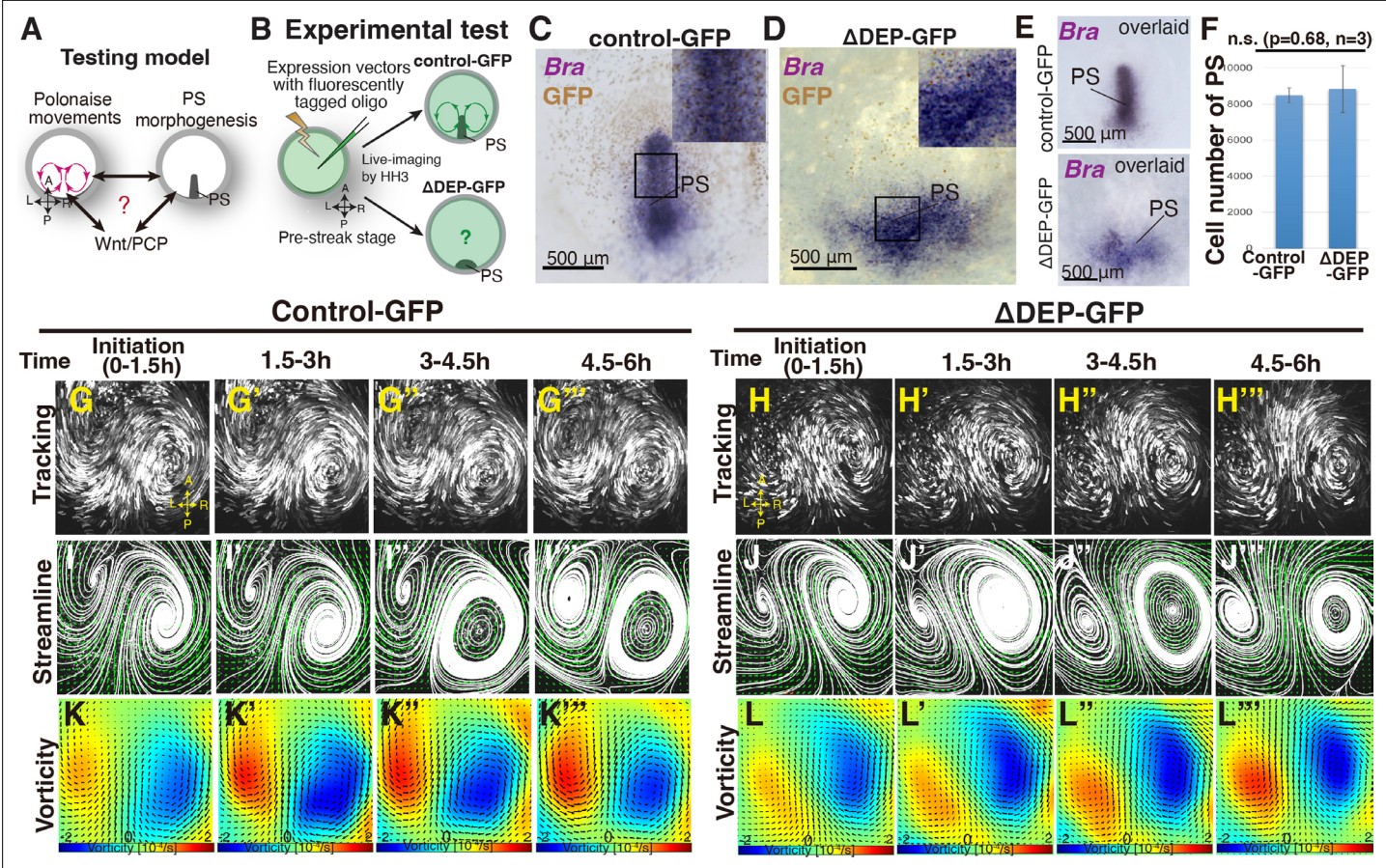

**Figure 1.** 'Polonaise movements' persist under suppression of Wnt/PCP pathway. (**A**) Testing model of relationships between the polonaise movements, primitive streak (PS) morphogenesis, and the Wnt/PCP pathway. A-P and L-R; anterior-posterior and left-right body axes, respectively. (**B**) Diagram of experimental set up. (**C, D**) In situ hybridizations for Brachyury (Bra) in control- and ΔDEP-GFP-misexpressing embryos. (**E**) Overlay of embryos hybridized for Bra (n=4 for each). (**F**) Cell number in PS [control 8.5±0.4 × 10³ cells vs ΔDEP 8.8±1.2 × 10³ cells, p=0.68, n=3 for each (two-tailed Student's t test)]. (**G-H'''**) Trace of cell flow path of electroplated epiblast cells with Flowtrace. The initiation of the polonaise movements was set as Time 0 (t=0). See also *Video 1* and *Figure 1—source data 1*. (**I-J'''**) Streamlines, visualizing averaged cell flows during each time period. (**K-L'''**) Vorticity plots, displaying an averaged measure of the local rotation during each time period. Blue, clockwise; red, counter-clockwise rotation. Scale bars: 500μm.

The online version of this article includes the following source data and figure supplement(s) for figure 1:

**Source data 1.** Vector field plots in control- and ΔDEP-GFP.

**Figure supplement 1.** Distance between the LR rotations increases under suppression of the Wnt/PCP pathway.

## Results

### Suppression of the Wnt/PCP pathway maintains the bilateral vortex-like counter-rotating cell flow along a deformed PS

Throughout this work, the cell flows were recorded by tagging individual epiblast cells with electroporated fluorescent tracers at the pre-streak stage (*Eyal-Giladi and Kochav, 1976*). The tagged embryos were live-imaged until HH3 with a modified New culture system, which we previously described (*New, 1955*; *Maya-Ramos and Mikawa, 2020*). To reconstruct trajectories of the tagged cells and quantitatively analyze the cell flow pattern, the recorded images were processed using tracking analysis tools (*Gilpin et al., 2017*), Particle Image Velocimetry (PIV) (*Thielicke and Stamhuis, 2014*; *Stamhuis, 2006*; *Raffel et al., 1998*; *Batchelor, 2000*; *Aris, 2012*), and subsequent visualization techniques (*Jiang et al., 2005*; *Greenshields and Christopher, 2022*).

To address the relationship between the polonaise movements and PS morphogenesis, we experimentally manipulated the shape of the PS while maintaining the population of PS cells (*Figure 1A–B*). The DEP domain of Dishevelled (Dsh; a transducer protein of Wnt signaling) is responsible for the

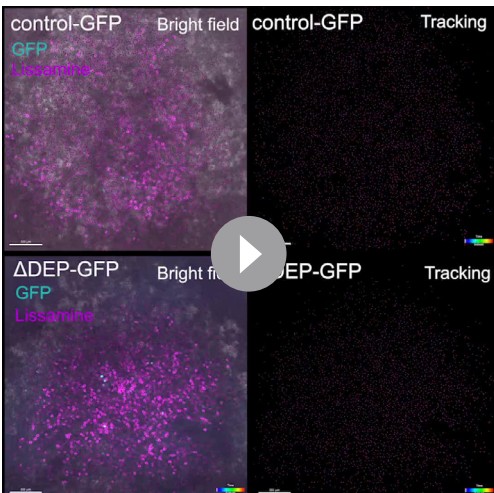

**Video 1.** 'Polonaise movements' continue under suppression of Wnt/PCP pathway. Time-lapse movie was analyzed by using Imaris tracking. Embryos were co-electroporated with lissamine-tagged oligo (red) and control-GFP or ΔDEP-GFP expression vectors (cyan) at pre-streak stage X-XII. Live imaging performed at 4 x objective lens on a spinning disk confocal microscope until HH3. Related to *Figure 1*, *Figure 1—figure supplement 1*, and *Figure 1—source data 1*. Scale bars, 300 μm.

https://elifesciences.org/articles/89948/figures#video1

non-canonical Wnt/PCP pathway (*Sharma et al., 2018*; *Rothbächer et al., 2000*), and misexpression of dominant-negative Dsh lacking DEP [dnDsh(ΔDEP)] leads to deformation of the midline structures, including the PS (*Voiculescu et al., 2007*). Further, the Wnt/PCP pathway is involved in cellular polarity and migration, while the canonical Wnt pathway regulates cell proliferation (*Yang and Mlodzik, 2015*). We refer the dnDsh(ΔDEP)-GFP construct that we generated, as ΔDEP-GFP, and tested its ability to alter cellular polarity, resulting in PS deformation' (*Figure 1C–E*, *Figure 1—figure supplement 1A–D*). In the control-GFP-introduced PS and epiblast cells, Pk1 (a PCP core component) exhibited a polarized localization at the E-cadherin-based adherens junctions, particularly cell-cell junctions (*Figure 1—figure supplement 1A–B*, n=3). In contrast, this Pk1-localization pattern was not identified in the ΔDEP-GFP-misexpressing PS and epiblast cells (*Figure 1—figure supplement 1C–D*, n=3), implying that the construct efficiently disrupted the cellular polarity within the misexpressing cells. The expression of *Brachyury* indicated that ΔDEP-GFP-PS cells maintained their identity (*Figure 1E*) but displayed a shorter and wider morphogenesis than the control PS (*Figure 1C–E*, n≥4 for each). The number of PS cells was not significantly different from the control- and ΔDEP-GFP-PS (control 8.5±0.4 × $10^3$ cells vs ΔDEP 8.8±1.2 × $10^3$ cells, p=0.68, n=3 for each, *Figure 1F*). These results indicated that ΔDEP-misexpression was capable of suppressing the Wnt/PCP pathway through disrupting the localization pattern of the PCP core component while maintaining the number of the PS cells. Having confirmed the function of the ΔDEP-GFP construct, we applied it as a molecular tool for manipulating the shape of the PS.

To address relationship(s) between the polonaise movements and the shape of the PS, we live-imaged the control- and ΔDEP-GFP-embryos from pre-streak stage to HH3 and analyzed the resulting cell flows (*Figure 1G–L"'*, and *Video 1*). In the control-GFP embryo, the cell flow pattern, which was visualized by tracking tools and quantitatively analyzed by PIV with subsequent flow-visualization techniques (*Jiang et al., 2005*; *Greenshields and Christopher, 2022*), identified robust polonaise movements (*Figure 1G–G"'I–I"' and K–K"'*, and *Video 1*, n=5). Surprisingly, these analyses indicated that suppression of the Wnt/PCP pathway maintained the polonaise movements despite PS deformation (n=5, *Figure 1C–L"'* and *Video 1*). While the topology of the counter-rotating flow pattern were maintained under suppression of the Wnt/PCP pathway (*Figure 1J–J'"*), the distance between the left-right rotating cell flows was wider in the ΔDEP embryos than the controls (control 860.2±86.2 μm vs ΔDEP 1008.4±183.8 μm, p=0.026, n=4 for each, *Figure 1—figure supplement 1E–I*), implying that the shape of the PS and/or the Wnt/PCP signaling-mediated cell intercalation affected the location of these rotations. These data demonstrated that the polonaise movements persisted even though the PS as a midline structure was deformed under the condition of suppressing the Wnt/PCP pathway.

## Mitotic arrest diminishes PS formation but initially preserves the bilateral vortex-like rotating cell flow

The above data showed that a deformed midline structure allowed the polonaise movements to occur while the position of the LR rotations in the polonaise movements was affected (*Figure 1* and *Figure 1—figure supplement 1*). However, it remained unclear the relationship between the polonaise movements and the population of the PS cells, particularly expansion of the PS precursor cells in PS

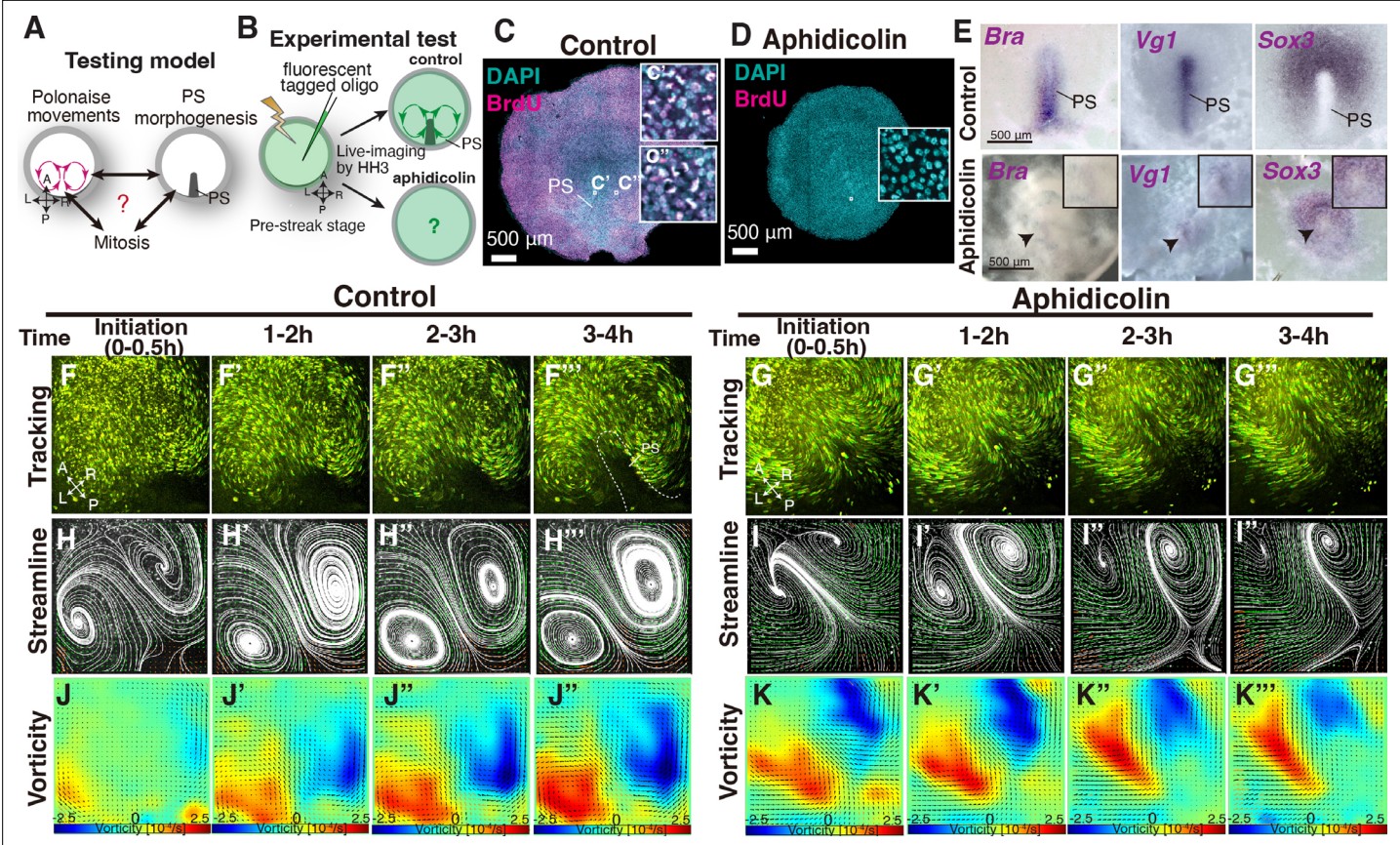

**Figure 2.** Early phase of 'polonaise movements' remains under global mitotic arrest. (**A**) Testing model of relationships between the polonaise movements, primitive streak (PS) morphogenesis, and mitosis. Axes as in *Figure 1*. (**B**) Diagram of experimental set up. (**C, D**) BrdU incorporation in control-sham operated and aphidicolin treated in embryos. White boxes show enlarged areas. (**C', C''**) High magnification of boxed areas in PS and non-PS, respectively. (**E**) In situ hybridizations for *Brachyury* (*Bra*), *Vg1*, and *Sox3* in control and aphidicolin-treated embryos. Arrow heads indicate diminished PSs. Black boxes, enlarged area including the diminished PS. (**F-G'''**) Trace of cell flow path with Flowtrace analysis (green is trace, yellow indicates endpoint) of fluorescently-tagged epiblast cells. The initiation of the polonaise movements was set as Time 0 (t=0). See also *Video 2* and *Figure 2—source data 1*. (**H-I'''**) Streamlines, visualizing averaged cell flows during each time period. Interpolated vectors are displayed in orange. (**J-K'''**) Vorticity plots, displaying averaged measure of the local rotation during each time period. Blue, clockwise; red, counter-clockwise rotation. Scale bars: 500µm. **Note**: Since vorticity is calculated for all deviation from set point, slight curves and full rotations receive the same color indication, as seen in K''' which is not maintaining a bilateral vortex-like-rotating cell flows. See Materials and methods section for a full discussion.

The online version of this article includes the following source data and figure supplement(s) for figure 2:

**Source data 1.** Vector field plots in control- and aphidicolin-treatment.

**Figure supplement 1.** PS extension dose-dependently responds to aphidicolin.

**Figure supplement 2.** Aphidicolin-treatment maintains apoptotic index and pMyosin cables.

morphogenesis (*Figure 2A–B*). To address this question, we applied a mitotic arrest approach using aphidicolin (a specific DNA polymerase inhibitor), which has been shown to diminished PS development (*Cui et al., 2005*; *Saadaoui et al., 2020*). In chick embryonic disc, active cell proliferation was identified in both the epiblast and PS cells (*Figure 2C*: *Sanders et al., 1993*). As verified by BrdU incorporation assay, the aphidicolin-treated embryos exhibited little or no mitotic activity (*Figure 2D*). Morphogenesis of the PS was monitored by *Brachyury* (a marker for PS and mesodermal cells) and *Sox3* (an epiblast marker). The expression domain and midline extension of cells, expressing *Brachyury* and negative for *Sox3*, were considerably reduced in the aphidicolin-treated embryos, compared to the control (*Figure 2E*; n≥4 for each). The expression pattern of *Brachyury* and PS extension followed an aphidicolin dose-dependent diminishment (*Figure 2—figure supplement 1A-D*). The *Vg1*-expressing area was also diminished but detectable in the aphidicolin-treated embryos (*Figure 2E*, n=4), suggesting that the axis-inducing morphogen was preserved in the embryonic disc, but expansion of

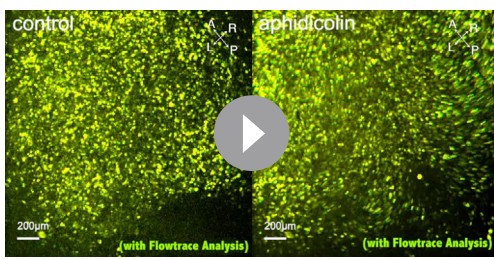

**Video 2.** Mitotic arrest maintains early phase of 'polonaise movements'. Time-lapse movie of fluorescently tagged epiblast cells in control and aphidicolin-treated embryos from pre-streak to HH3, processed by using Flowtrace. The trajectories, from green to yellow, indicate movement of fluorescently tagged cells (30 min projections). Embryos were electroporated with fluorescein-tagged oligo (green) and treated with DMSO (control-sham operated) or 100 µM of aphidicolin. Live-imaging performed at 4 x objective lends on a spinning disk confocal microscope for 10 hr. Related to *Figure 2*, *Figure 2—figure supplement 2*, *Figure 2—figure supplement 2*, and *Figure 2—source data 1*. Scale bars, 200 µm.
https://elifesciences.org/articles/89948/figures#video2

the PS precursor cells was suppressed by mitotic arrest. These results indicate that mitotic arrest, induced by the aphidicolin-treatment, remarkably reduced PS extension and development, consistent with the previous works (*Cui et al., 2005*; *Saadaoui et al., 2020*).

To address the relationship between the polonaise movements and the population of the PS cells, we live-imaged the fluorescently tagged embryos under aphidicolin-treatment and analyzed the resulting cell flow (*Figure 2F–K'''* and *Video 2*). Tracking and PIV-based analyses visualized the cell flow pattern in control and aphidicolin-treated embryos (*Figure 2F–K'''* and *Video 2*). In control embryos, the polonaise movements occurred from pre-streak until HH3 for more than four hours (n=4, *Figure 2F–F'''* and *Video 2*). The streamline and vorticity plots illustrated that the polonaise movements became more robust in a time-dependent manner (*Figure 2H–H'''* and *J–J'''*). Strikingly, in the aphidicolin-treated embryos (n=4), the polonaise movements initiated but were maintained for a shorter period, 2.2±1.2 hr (control 6.5±1.2 hr, p=0.0038, Figure G-G'", I-I'", and *Video 2*). The streamline and vorticity plots also indicated the initiation and early phase of the polonaise movements in the aphidicolin-treated embryos (*Figure 2J–J'''* and *K–K'''*). Together, these data demonstrated that the typical pattern of the polonaise movements was induced and initially maintained in the short-term, while there was no or little indication of robust PS formation or extension.

Aphidicolin potentially induces apoptosis (*Glynn et al., 1992*), which might indirectly lead to the cell flow changes through tissue tension alterations. To test this possibility, we examined apoptosis in the control and aphidicolin-treated embryos (*Figure 2—figure supplement 2A-B*). TUNEL assay showed that the aphidicolin-treatment slightly increased apoptotic rate but there was no significant difference between these embryos and controls (control sham, 10.6 ± 3.5%; aphidicolin, 12.1 ± 3.7%, p=0.20, n=4 for each, *Figure 2—figure supplement 2B*), suggesting that the bilateral rotating cell flow in the aphidicolin-treated embryos was not indirectly induced by apoptosis.

Previous studies have shown that a large-scale ring-like distribution of phospholyrated-myosin (pMyosin) cables among the epiblast cell layer is required for tissue tension related to cell flow (*Saadaoui et al., 2020*; *Chuai et al., 2023*; *Rozbicki et al., 2015*). We examined whether the pMyosin cables were maintained in the treated embryos over the course of the imaging (*Figure 2—figure supplement 2C-D"*). In the aphidicolin-treated embryos, the large-scale ring-like distribution pattern of the pMyosin cables remained in the epiblast (*Figure 2—figure supplement 2D-D"*), similar to the control embryos (*Figure 2—figure supplement 2C-C"*), implying that the mitotic arrested-embryos maintained tissue tension, consistent with the previous report (*Saadaoui et al., 2020*). Thus, these results support a model that mitosis is coupled with PS morphogenesis but is not necessarily required for the initiation and early phase of the polonaise movements.

## PS properly extends along the midline despite altering the large-scale cell flow

Given that PS morphogenesis is required for maintenance of the polonaise movements, we next examined the converse of the relationship, that the polonaise movements may be required for PS formation. To test this model, we experimentally reprogrammed the large-scale cell flow (*Figure 3A–B*). Ectopically induced Vg1 leads to formation of a *Brachyury*-positive secondary PS and a secondary midline axis (*Figure 3C–D*: *Cooke, 1973*; *Wei and Mikawa, 2000*; *Shah et al., 1997*). COS cells expressing either a control-GFP (control-GFP/COS) or Vg1 (Vg1/COS) construct were placed at the

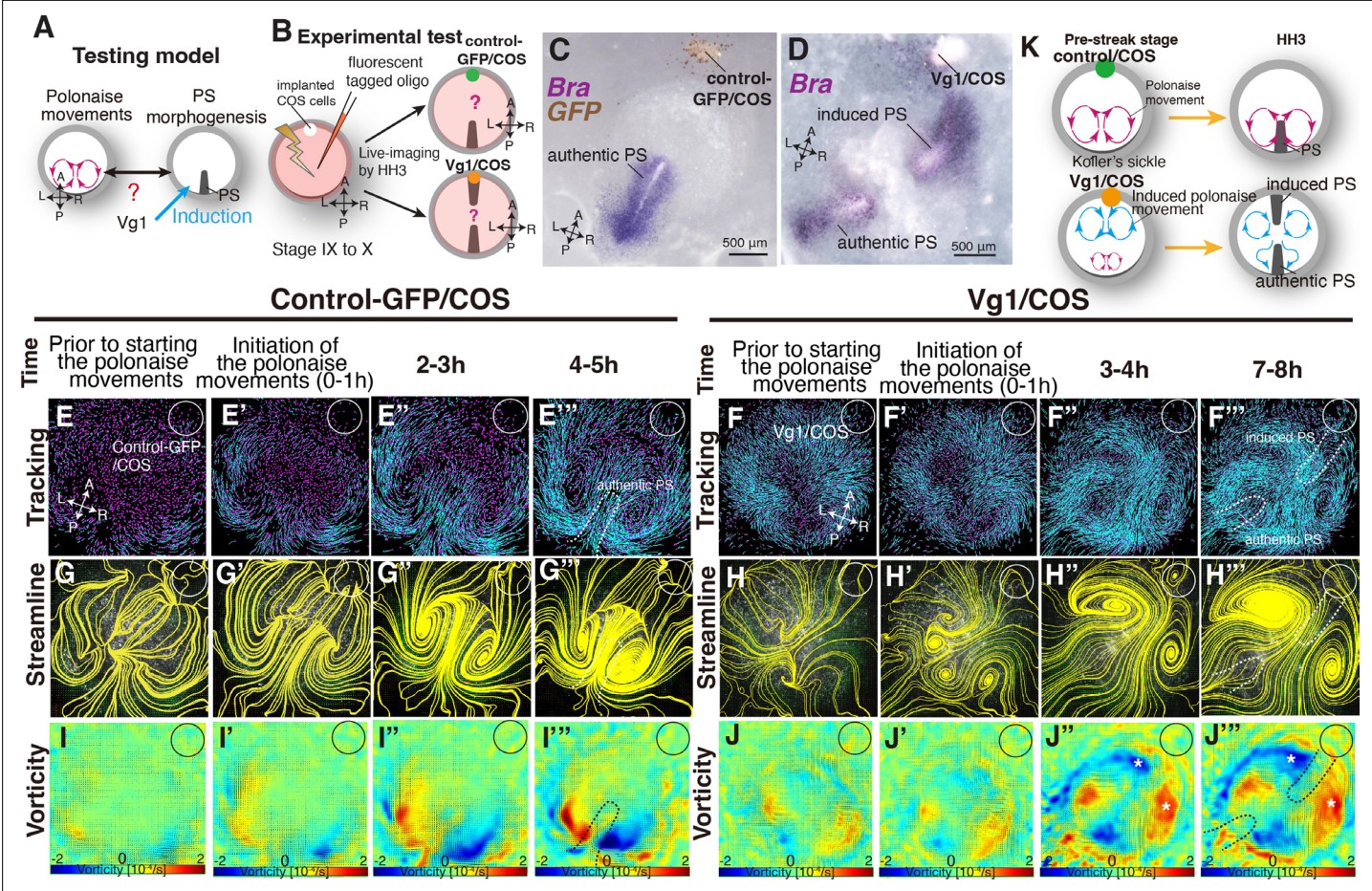

**Figure 3.** Authentic PS extends under disruption of authentic 'polonaise movements'. (**A**) Testing model of relationships between the polonaise movements, primitive streak (PS) morphogenesis, using ectopic Vg1-inducing system. Axes as in *Figure 1*. (**B**) Diagram of experimental set up. (**C, D**) In situ hybridizations for Brachyury (Bra) in control and ectopic Vg1-induced PS, respectively. (**E-F'''**) Trace of cell flow path of electroplated epiblast cells by using Imaris tracking analysis. See also *Video 3* and *Figure 3—source data 1*. White circles, COS cell implanted site. The initiation of the polonaise movements was set as Time 0 (t=0). (**G-H'''**) Streamlines, visualizing averaged cell flows during each time period of the vector fields. (**I-J'''**) Vorticity plots, displaying an averaged measure of the local rotation during each time period in the vector fields. Blue, clockwise; red, counter-clockwise rotation. Scale bars: 500μm. **Note**: vorticity is identified in both a full-rotation and curve, particularly in (**J-J'''**). The white asterisks indicate the polonaise movements at the induced PS which are opposite direction to the authentic PS. (**K**) Summary of the cell flow patterns in control-GFP/COS- and Vg1/COS-implanted embryos, shown in (**E-J'''**).

The online version of this article includes the following source data and figure supplement(s) for figure 3:

**Source data 1.** Vector field plots in control-GFP/COS and Vg1/COS.

**Figure supplement 1.** Time evolution of cell flow in control-GFP/COS-implanted embryo.

**Figure supplement 2.** Time evolution of cell flow in Vg1/COS-implanted embryo.

anterior marginal zone (AMZ) at pre-streak stage, approximately 180° from the posterior marginal zone where the authentic PS forms, and we analyzed the resulting cell flow during both authentic and ectopically induced PS extension (*Figure 3B*). If the bilateral rotating cell flow is required for PS formation, both the authentic and induced PSs should each be accompanied by polonaise movements during the extension. The trajectory analysis by Imaris and PIV-based analyses showed that the control-COS implanted embryos exhibited stereotypical polonaise movements along the authentic midline axis, indicating that the COS cell implantation maintained an authentic cell flow pattern (*Figure 3E–E'''*, *G–G'''I–I'''* and K, Fig. S4, and *Video 3*, n=4). In contrast, the analyses of the cell flow in the Vg1/COS implanted embryos illustrated the disruption of the authentic polonaise movements at the extending authentic PS (*Figure 3F–F'''*, *H–H'''* and J–J''', Fig. S5, and *Video 3*, n=4). At pre-streak stage, the epiblast cells initially converged to the center of the embryonic disc along both the

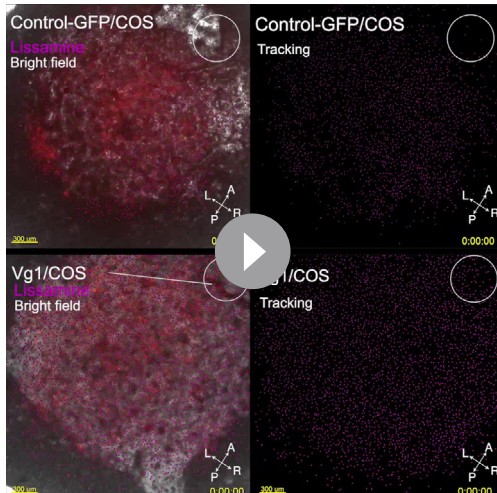

**Video 3.** Authentic PS extends in absence of regular 'polonaise movements'. Embryos were electroporated with lissamine-tagged oligo (red) and implanted with COS cells expressing either a control-GFP (control-GFP/COS) or Vg1 (Vg1/COS) expression vector at anterior marginal zone at pre-streak stage IX-X. Live-imaging performed at a 2 x objective lens on a wide-field epifluorescent microscope. Magenta dots; lissamine-tagged epiblast cells, cyan lines; trajectories of the tagged epiblast cells for 2 hours. Related to *Figure 3*, *Figure 3—figure supplement 1*, *Figure 3—figure supplement 2*, and *Figure 3—source data 1*. Scale bars, 300 μm.

https://elifesciences.org/articles/89948/figures#video3

authentic and the induced axes, and the bilateral rotating cell flows began to appear at both axes (*Figure 3F–F'''' and H–H'''*, Fig. S5, and *Video 3*). This cell flow pattern continued for approximately one hour (*Figure 3H'*, Fig. S5, and *Video 3*). Before extension of the authentic PS, the bilateral rotating cell flow along the induced axis became dominant, and eventually overrode the original cell flow (*Figure 3H'–H''''*, Fig. S5, and *Video 3*). Despite global disruption and reversed counter-rotating cell flows, the authentic PS extended properly along the authentic midline (*Figure 3D, F'''', H''', J''' and K*, Fig. S5, and *Video 3*; n=4). Secondary midline-axis induction by ectopic Vg1 created the polonaise movements around the secondary PS and altered the original cell flow pattern, generating a reprogrammed large-scale cell flow (*Figure 3F–F''''*, *H–H''''*, *J–J''''* and K, Fig. S6, and *Video 3*). These results indicate that extension of the authentic PS is preserved even under the defective polonaise movements.

## Vg1 is capable of inducing the bilateral vortex-like rotating cell flow despite diminished PS extension under mitotic arrest

The above data indicate the lack of a strong relationship between the polonaise movements and PS morphogenesis; however, it remained unclear how the polonaise movements were induced. Given that mitotic arrest maintained both Vg1-expression and the initiation of the polonaise movements (*Figure 2* and *Video 2*) and that induction of the secondary midline axis by ectopic Vg1 reprograms the epiblast cell flow (*Figure 3*, *Figure 3—figure supplement 1*, *Figure 3—figure supplement 2*, and *Video 3*), we tested the possibility that the axis-inducing morphogen, Vg1, would be capable of inducing the polonaise movements in the absence of PS extension (*Figure 4A*). We combined the secondary axis induction by AMZ implantation of Vg1/COS cells with aphidicolin-treatment and analyzed the resulting cell flow (*Figure 4B*, *Figure 4—figure supplement 1*, *Figure 4—figure supplement 2*, and *Video 4*). The BrdU assay showed that the aphidicolin treatment effectively blocked mitosis in Vg1/COS implanted embryos (control sham 88±9.0% vs control-GFP/COS with aphidicolin 0.2±0.1% vs Vg1/COS with aphidicolin 0.19 ± 0.2%, p=0.0005 × 10$^{-3}$, n=3 for each, *Figure 4C–D*) and diminished both the authentic and induced PSs (*Figure 4E–F*, n=3 for each). The control-GFP/COS implanted embryo under mitotic arrest condition initially displayed the polonaise movements along the authentic midline axis, however, the vorticity decreased time-dependently, resulting in a defective rotating cell flow (n=3, *Figure 4G–G'''I–I'''* and *K–K'''*, *Figure 4—figure supplement 1*, and *Video 4*), consistent with the non-COS cell implanted embryos under aphidicolin-treatment (shown in *Figure 2*). In contrast, the Vg1/COS implanted embryos under mitotic arrest condition exhibited the polonaise movements predominantly at the induction point of the secondary axis with disrupted authentic cell flow at the site of the authentic midline axis (n=3, *Figure 4H–H'''*, *J–J'''* and *L–L'''*, and *Video 4*). These data suggest that the midline axis induction by the single morphogen brings about the polonaise movements, even when the midline structure (i.e. extending PS) is diminished. Taken together, our data show that Vg1 is capable of inducing and reprogramming the large-scale cell flow during gastrulation even though the midline structure is diminished.

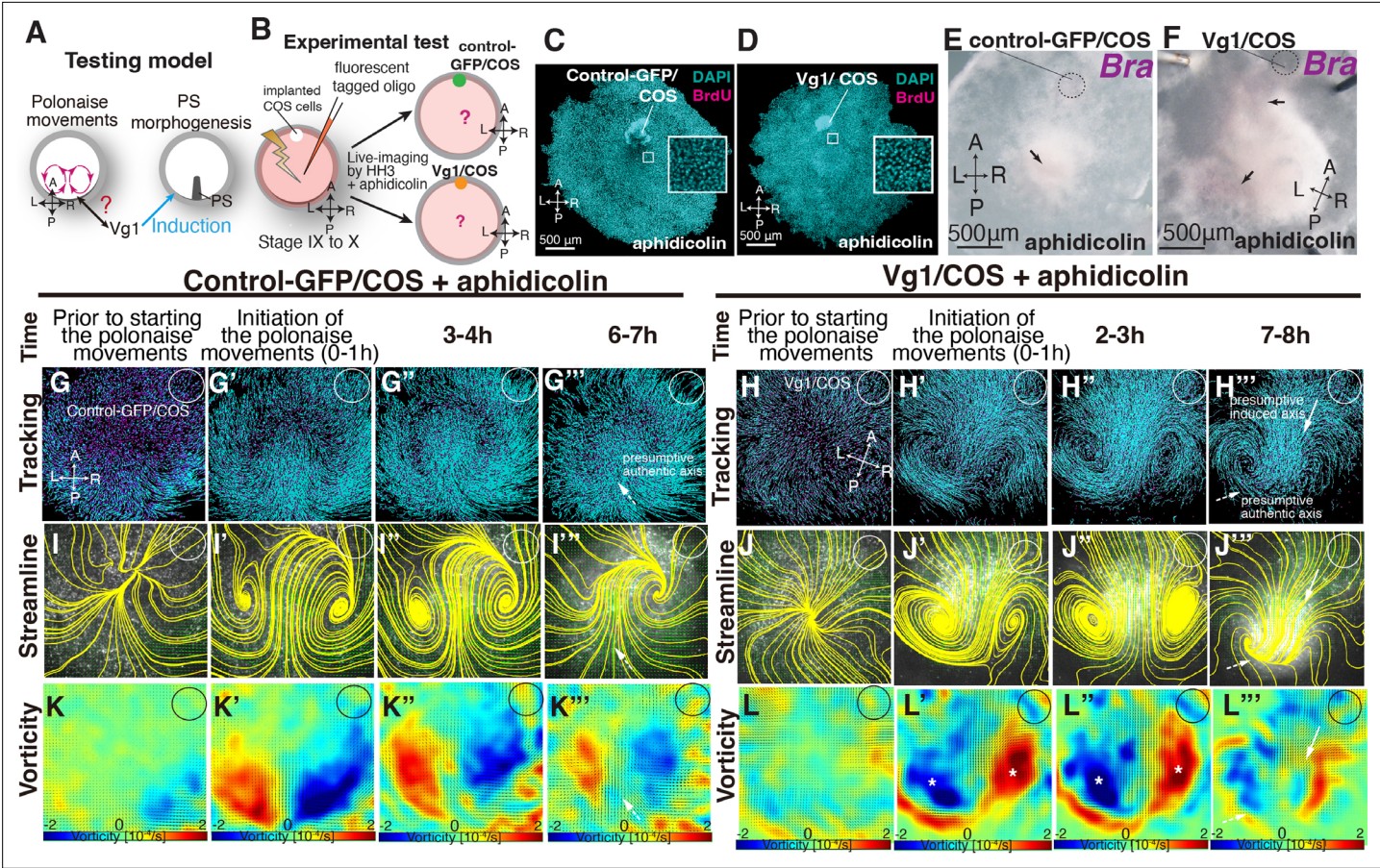

**Figure 4.** Induced 'polonaise movements' persist despite defective PS morphogenesis. (**A**) Testing model of relationships between the polonaise movements, primitive streak (PS) morphogenesis, and Vg1. Axes as in *Figure 1*. (**B**) Diagram of experimental set up. (**C, D**) BrdU incorporation in control-GFP/COS- and Vg1/COS-implanted embryos under aphidicolin-treatment. White boxes showed enlarged areas. (**E, F**) In situ hybridizations for Brachyury (Bra, n=3 for each). Black arrows indicate diminished PSs. (**G-H'''**) Trace of cell flow path of electroplated epiblast cells by using Imaris tracking analysis. See also *Video 4* and *Figure 4—source data 1*. White circles, COS cell implanted site. The initiation of the polonaise movements was set as Time 0 (t=0). (**I-J'''**) Streamlines, visualizing averaged cell flows during each time period of the vector fields. (**K-L'''**) Vorticity plots, displaying an averaged measure of the local rotation during each time period. Note: vorticity is identified in both a full-rotation and curve, particularly in (**L-L'''**). Blue, clockwise; red, counter-clockwise rotation. White asterisks, the polonaise movements at the induced axis which are opposite direction to the authentic midline axis. Scale bars: 500µm. **Note:** vorticity is identified in both a full-rotation and curve in (**K'''**) and (**L'''**). See description in the Materials and methods section.

The online version of this article includes the following source data and figure supplement(s) for figure 4:

**Source data 1.** Vector field plots in control-GFP/COS and Vg1/COS under aphidicolin-treatment.

**Figure supplement 1.** Time evolution of cell flow in control-GFP/COS-implanted embryo under aphidicolin-treatment.

**Figure supplement 2.** Time evolution of cell flow in Vg1/COS-implanted embryo under aphidicolin-treatment.

## Discussion

Here, we described a previously undefined relationship between the polonaise movements and PS morphogenesis (*Figure 5*). Our approaches to molecularly manipulating PS morphogenesis revealed that the initiation and early phase of the polonaise movements occur despite diminished PS formation (*Figure 5A*). Conversely, the experimental manipulation of the cell flow demonstrated that the authentic PS extends regardless of aberrant flow patterns (*Figure 5B*). Further, a single axis-inducing morphogen, Vg1, is capable of inducing the bilateral rotating cell flow even under mitotic arrest, when the ectopic PS is reduced (*Figure 5B and C*). Our data support a model in which the PS population is required for maintaining the polonaise movements, as seen in the ΔDEP embryos, while the polonaise movements are not necessarily responsible for PS morphogenesis (*Figure 5D*).

A similar mitotic arrest has been described as simultaneously diminishing PS extension and disrupting the polonaise movements (*Cui et al., 2005*; *Saadaoui et al., 2020*), but it was not

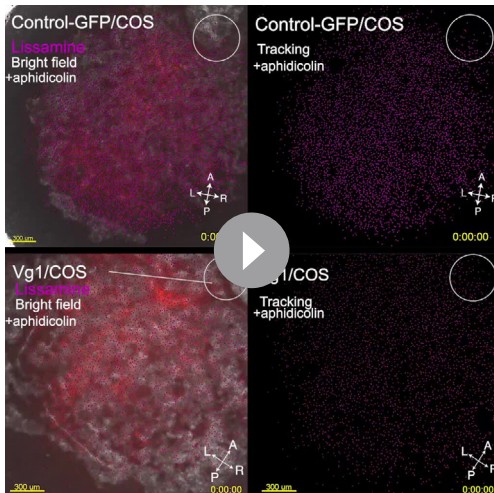

**Video 4.** Ectopic Vg1 leads to 'polonaise movements' despite defective PS morphogenesis after mitotic arrest. Embryos were electroporated with lissamine-tagged oligo (red), implanted with either control-GFP/COS or Vg1/COS at the anterior marginal zone (AMZ), and treated with 100 μM of aphidicolin at pre-streak stage IX to X. Live imaging performed at a 2 x objective lens with a wide-field epifluorescent microscope. Magenta dots; lissamine tagged epiblast cells, cyan lines; trajectories for 2.5 hr. Related to *Figure 4*, *Figure 4—figure supplement 1*, *Figure 4—figure supplement 2*, and *Figure 4—source data 1*. Scale bars, 300 μm.

https://elifesciences.org/articles/89948/figures#video4

determined whether a lack of PS extension and/or diminished cell population of the PS at the midline led to defective polonaise movements. In this study, our live-imaging system combined with the quantitative analyses, molecular tools for manipulating PS formation, and the gene expression analyses allowed us to clarify the relationships among PS morphogenesis, the polonaise movements, mitosis, and the Wnt/PCP pathway (*Figure 1*, *Figure 2*, and *Figure 5A and C*). Our data demonstrated that the shape of the PS as a midline structure is not necessarily responsible for initiation of the polonaise movements while the population of the PS cells, expanded by mitosis, is required for sustained polonaise movements (*Figure 1* and *Figure 2*). Mesendoderm specification is required for large-scale cell flows associated with later stage ingression and intercalation at the PS during gastrulation (*Chuai et al., 2023*; *Serrano Nájera and Weijer, 2023*). Here, we have described the distinctive set of cell flows that precede gastrulation in the chick embryo, and the relationship between early and late cell flow has yet to be examined.

The Wnt/PCP pathway has been assumed to be coupled with the polonaise movements, as an evolutionarily conserved mechanism (*Voiculescu et al., 2007*; *Roszko et al., 2009*). In contrast, another study showed that misexpression of dnWnt11 or dnDsh-ΔPDZ [suppressing both the Wnt canonical and non-canonical pathway, including the Wnt/PCP (*Sharma et al., 2018*)] maintained the bilateral counter-rotating cell flow along a defective PS (*Chuai et al., 2006*). Thus, the role of the Wnt/PCP signaling pathway for the polonaise movements has remained unresolved. In this study, our loss-of-function technique, introducing ΔDEP to PS cells, demonstrated that the polonaise movements were preserved along the deformed PS (*Figure 1*, and *Video 1*). Our electroporation-mediated gene-transfection allowed us to restrict gene transduction in area and number of cells (*Ishii and Mikawa, 2005*). Despite the technical limitations, the suppression level of the Wnt/PCP pathway through the ΔDEP-misexpression was sufficient to lead to the PS deformation (*Figure 1* and *Figure 1—figure supplement 1*). The Wnt/PCP pathway potentially works both cell-autonomously and non-cell autonomously (*Yang and Mlodzik, 2015*; *Davey and Moens, 2017*). In the PS and non-PS epiblast cells, the disruption of the Pk1-localization was identified not only in the ΔDEP-misexpressing cells, but also in neighbors; however, this study is not designed to provide insight into questions of cell autonomy. Further, the Wnt/PCP pathway is a major regulator of cell intercalation (*Roszko et al., 2009*; *Tada and Heisenberg, 2012*). Recent studies have shown that cell intercalation plays an important role in PS morphogenesis and the initiation of the polonaise movements (*Saadaoui et al., 2020*; *Voiculescu et al., 2007*; *Chuai et al., 2023*; *Rozbicki et al., 2015*; *Voiculescu et al., 2014*). Our data demonstrates that suppression of the Wnt/PCP pathway preserves the initiation and maintenance of the polonaise movements, but the lateral positioning of the rotations is enlarged when the PS is deformed (*Figure 1* and *Figure 1—figure supplement 1*). These results suggest that the Wnt/PCP pathway may not be associated with directional cell intercalation for driving the polonaise movements (*Figure 5C*).

To investigate the role of the polonaise movements for PS morphogenesis, a technical challenge/limitation is how to stop or disrupt the cell flow while preserving the cellular process intact. For example, previous loss-of-function studies, using knockdown techniques (such as siRNA and

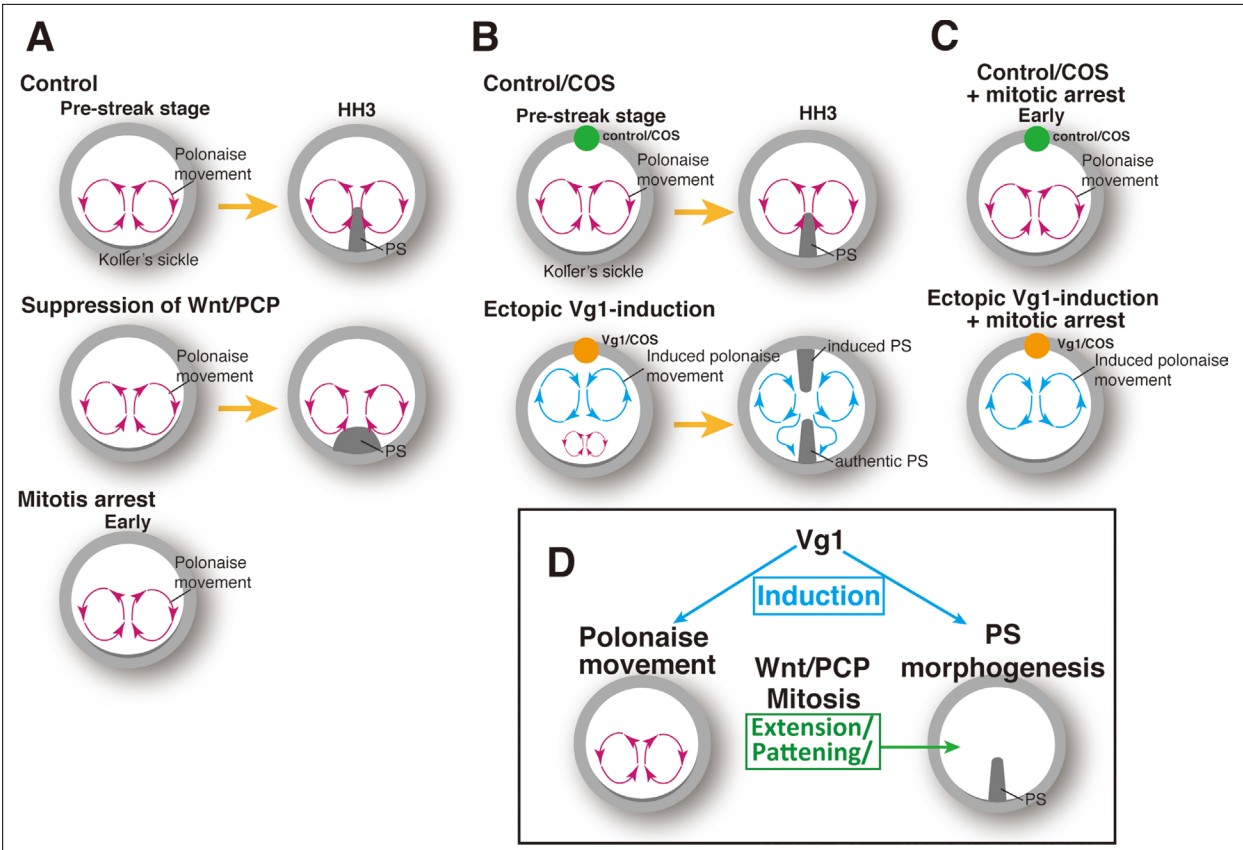

**Figure 5.** Summary of the patterns of the cell flow and PS morphogenesis. (**A**) Cell flow pattern under experimental manipulation of PS morphogenesis. (**B**) Authentic PS morphogenesis under disrupting authentic polonaise movements. (**C**) Vg1-induced cell flow pattern under defective PS morphogenesis. (**D**) Summary of the relationships between polonaise movements, PS morphogenesis, mitosis, and Wnt/PCP pathway.

inhibitor treatments), identified the necessity of myosins, including pMyosin and actomyosin, for both the large-scale cell flow and PS morphogenesis, through regulation of tissue tension/biomechanical force (*Chuai et al., 2023*; *Rozbicki et al., 2015*). Myosins are fundamental cellular components and involved in several cell biological processes, such as tissue tension, cell division, and intracellular transport (*Hartman and Spudich, 2012*). Therefore, it remains a possibility that myosin-related mechanisms could be involved in the cell flow and PS morphogenesis. Recent studies using injury and pressure models further demonstrated that the cell flow pattern can be altered by a physical cut to an embryonic disc or removal of a large portion of an embryo disc through change of the tissue tension or ablation (*Saadaoui et al., 2020*; *Caldarelli et al., 2021*). While these models suggest that the cell flow pattern is dependent on an intact embryonic architecture, it remains unclear as to involvement of other elements such as signaling pathways and gene expression, which can be also molecularly and cell-biologically changed as a result of surgeries. Thus, we developed an experimentally manipulation of the cell flow system, using a Vg1-induced two-midline axis model, disturbing the stereotypical cell flow pattern while maintaining embryonic development (*Figure 3* and *Figure 5B*).

Our two-midline axis model provided an important insight into the axis-inducing morphogen for induction and patterning of the polonaise movements (*Figure 3*, *Figure 4*, *Figure 3—figure supplement 1*, *Figure 3—figure supplement 2*, *Figure 4—figure supplement 1*, *Figure 4—figure supplement 2*, *Video 3*, and *Video 4*). Our data demonstrated that the ectopic Vg1, mimicking the inducing activity of the posterior marginal zone, generated the bilateral rotating cell flow aligned to the induced midline axis but disrupted the cell flows at the authentic midline axis (*Figure 3* and *Figure 4*). Despite the disturbed cell flow, PS extension was preserved along both authentic and induced midline axes (*Figure 4*). While this model ultimately disturbed the stereotypical cell flow pattern at the authentic midline axis, cell flow was initiated there. Therefore, our study does not rule out potential roles of the cell flow in the initiating steps of PS extension.

It remains unclear how the induced polonaise movements along with the secondary axis overcame the authentic cell flows despite mitotic arrest (*Figure 4* and *Figure 5*). As for axis-induction by other axis-inducing morphogens (*Green, 2002*), Vg1 may work dose-dependently for induction and patterning of the polonaise movements, and ectopic application may have over-ridden endogenous levels (*Figure 3* and *Figure 4*). Further studies will clarify the relationship between the large-scale cell flows and the concentration of the morphogen; this will add new insight to the regulation mechanisms that connect axis-induction and the cell flow patterning.

This study further suggests evolutional distinctions underlying cell biological mechanisms of the large-scale cell flow and midline morphogenesis, between amniotes and non-amniotes. The first distinction is the necessity of the Wnt/PCP pathway to the large-scale cell flow. Whereas the Wnt/PCP pathway is coupled with both the large-scale cell flow and midline morphogenesis in non-amniotes (*Tada and Heisenberg, 2012*), our data revealed that the signaling pathway is required for proper PS morphogenesis, but contributes little to the polonaise movements (*Figure 1*). The second distinction is mitosis-dependency/independency comparisons between model systems of gastrulation (*Figure 2*). Non-amniotes, in particular amphibians, utilize a blastopore-mediated gastrulation and undergo notochord formation with the large-scale cell flow, all independent of mitosis (*Keller et al., 2000*). In amniotes, mitosis is required for PS morphogenesis, and expanding the mass of the PS cells contributes to the maintenance of the polonaise movements (*Figure 2*). Thus, the large-scale cell flow and midline morphogenesis in amniote gastrulation is not tightly coupled, which may be evolutionarily distinct from non-amniote gastrulation.

In summary, our study illustrates the relationships between midline morphogenesis, the large-scale cell flow, and an axis-inducing factor in amniote gastrulation. These findings will add another layer of our understanding to how the bilateral body plan is initiated and patterned along the midline axis.

## Materials and methods

### Embryo isolation and culture conditions

Fertilized eggs of White Leghorn (*Gallus Gallus domesticus*) were obtained from Petaluma Farms (Petaluma, CA) and were incubated at 37 °C in a humidified incubator to the appropriate embryonic stages. We electroporated unincubated eggs at pre-streak stages IX to XII for cell flow imaging. Embryos were isolated in Tyrode's solution (137 mM NaCl, 2.7 mM KCL, 1 mM MgCl2, 1.8 mM CaCl2, 0.2 mM Ha2HPO4, 5.5 mM D-glucose, pH 7.4). After manipulations (such as electroporation, inhibitor treatment, and implantation), the embryos were live-imaged on a vitelline membrane stretched around a glass ring according to the New culture method as previously described (*New, 1955*; *Maya-Ramos and Mikawa, 2020*). All experiments were repeated more than three times, which were biological replicates.

### Implantation of COS cells

A total of $2x10^5$ of COS7 cells (ATCC, CRL-1651, this cell line was not listed in the ICLAC Register of Misidentified Cell Lines and negative for mycoplasma contamination) were plated in a 35 mm cell culture dish (Thermo Fisher Scientific, 153066), 24 hr prior to transfection. The cells were transfected by Lipofectamine 3000 (Thermo Fisher Scientific, L3000015) with 6 μg of pMT23-Vg1-myc-GDF1 plasmid DNA (a gift from Drs. Claudio D. Stern and Jane Dodd) for 5 hr. The next day, the transfected cells were trypsinized, and cultured using the hanging-drop method. Each hanging drop contained $10x10^6$ cells in 20 μl of culture media and were cultured overnight (O/N). The cell spheres were then rinsed in serum-free DMEM and implanted into the anterior marginal zone in the chick embryonic discs at pre-streak stages.

### Plasmid generation

ΔDEP-GFP was cloned from a plasmid, XE124 XDsh delta DEP-GFP-CS2+in pCS2+ (a gift from Dr. Randall Moon, Addgene, #16785), by PCR (Q5 High-Fidelity DNA Polymerase, New England Biolabs, M0491) with the following primers: 5'-TACCGCGGGCCCGGGATCCAGCCACCATGGCGGAGACT-3' and 5'- AGCCTGCACCTGAGGAGTGCTTATTTGTATAGTTCATCCATGCCATGTGTAATCC'. After PCR purification with QIAquick Gel Extraction Kit (Qiagen, #28704), the PCR product was inserted into a

backbone vector, pCAG-GFP (a gift from Connie Cepko, Addgene, #1115) by using Gibson-assembly (New England Biolabs, E2611).

## Electroporation

Embryos were transfected with expression vectors, and/or control-oligo DNA conjugated with Carboxyfluorescein or Lissamine (GENE TOOLS) using an electroporator (Nepagene) with 3 pulses of 2.4–3.8 Volts, 50 ms duration, 500 ms interval, and platinum electrodes. The DNA solution delivered to the epiblast contained 0.1% fast green (final 0.02%), 80% glucose (final 4%) and 5 µg/µl of expression vectors, or 1 mM control-oligo (5'-CCTCTTACCTCAGTTACAATTTATA-3').

## Aphidicolin treatment

Aphidicolin (SIGMA, A0781), dissolved in DMSO (SIGMA, B23151), was added to Tyrode's solution to a final concentration of 0.1–100 µM. Embryos were isolated, and soaked in Tyrode's solution with either 0.3% DMSO (control), or various concentrations of aphidicolin for 15 minutes at 37 °C. Embryos were then cultured using the New culture method at 37 °C.

## BrdU assay

Embryos were soaked in Tyrode's solution containing BrdU (final concentration; 0.1 mM, Thermo Fisher, B23151) with or without aphidicolin (SIGMA, A0781) at 37 °C for 15 min, and cultured for 12 hr at 37 °C (BrdU was incorporated for 12 hr). Embryos were then fixed in 4% paraformaldehyde (PFA, Electron Microscopy Sciences)/PBS for 30 min at room temperature (RT). Embryos were then washed with PBS to remove PFA, and unincorporated BrdU, and incubated with 1 M HCl for 1 hr at RT to denature DNA. BrdU signal was detected by immunofluorescence staining with anti-BrdU antibody (1:200, Millipore, MAB3424).

## Tunel assay

The In Situ Cell Death Detection Kit, TMR red (Rosche, 12156792910) was used, and the kit instructions followed with these additional modifications. Embryos were fixed in 4% PFA for 30 min at RT and washed in PBS 3 X for 10 min each. Permeabilization was performed in TBST (0.5% Triton-X) for 30 min at RT. Embryos were then incubated with the TUNEL reaction and DAPI staining for 3 hr at 37 °C and washed with PBS five times for 10 min each.

## Immunofluorescent staining

### For chicken embryos

Embryos were fixed in 4% PFA/PBS for 30 min at RT, or 4:1 Methanol/DMSO at 4 °C O/N and washed in PBS 3 X for 30 min each. Embryos were then incubated in blocking reagent [1% bovine serum albumin (BSA) and 0.1% triton in PBS] for 1 hr at RT. Embryos were then incubated with primary antibodies at 4 °C O/N. After washing in PBS 3 X for 30 min each, embryos were then incubated with secondary antibodies for 2 hr at RT. Embryos were then washed in PBS 2 X for 30 min each. For the BrdU assay, embryos had an additional incubation with Streptavidin, Alexa Fluor 647 conjugate (1:1000, Thermo Fisher Scientific) for 1 hr at RT, and washed in PBS 2 X for 30 min each. Embryos were incubated with DAPI for 1 hr, washed for 30 min, then mounted between cover slips (VWR, 48366249), and slide glasses (Thermo Fisher Scientific, 12-544-7) with Aqua-PolyMount (Polysciences, Inc, #18606–20).

### Primary antibodies

Anti-BrdU (1:200, Millipore, MAB3424), anti-E-Cadherin (1:500, BD Biosciences, 610181), anti-GFP (1:1000, Rockland, 600-101-215), anti-phospho myosin light chain 2 (1:300, Cell Signaling Technologies, 3674), anti-Prickle1 antibody (1:300, Proteintech, 22589–1-AP), and anti-GFP-HRP antibody (1:200, Abcam, 600-101-215), anti-ZO1 antibody (1:300, Thermo Fisher Scientific, 33–9100).

### Secondary antibodies

Anti-mouse-IgG (H+L) Alexa Fluor 647 antibody (1:500, Thermo Fisher Scientific, A-31571), anti-goat IgG (H+L) Alexa Fluor 488 (1:500, Thermo Fisher Scientific, A11055), anti-rabbit IgG (H+L) Alexa Fluor 568 antibodies (1:500, Thermo Fisher Scientific, A10042), anti-rabbit IgG (H+L) Alexa Fluor 647

(1:1000, Thermo Fisher Scientific, A-31573), anti-goat IgG Alexa Fluor 488 (1:1000, Thermo Fisher Scientific, A11055) and anti-mouse IgG Alexa Fluor 594 antibodies (1:1000, Thermo Fisher Scientific, A21203).

### Whole Mount In Situ Hybridization (WISH) and subsequent immunohistochemistry

Embryos were fixed in 4% PFA/PBS in PBS O/N at 4 °C. WISH was performed as previously described (*Batchelor, 2000*), using plasmids for *Brachyury* and *Sox3* (gift from Dr. Raymond B. Runyan), and *Vg1*. All WISH were carried out with paralleled control embryos for managing the color development time. For subsequent IHC, embryos were fixed in 4:1 methanol/DMSO at 4 °C O/N.

The next day, embryos were incubated with 4:1:1 methanol/DMSO/30% Hydrogen peroxide at RT and incubated for 2 hr. Then embryos were washed in Tris-Buffered Saline (pH7.6) with 0.1% Tween 20 (TBST) 3 X for 10 min each, and incubated with blocking solution (1% sheep serum in TBST) for 1 hr at RT. Primary antibodies, anti-GFP antibody-HRP conjugated (1:200, Abcam, ab6663), were applied for at 4 °C O/N incubation. Embryos were then washed in TBST 5 X for 1 hr each at RT, followed by 10 min with 0.05 M Tris-HCl (pH 7.6), and incubated for 30 min with DAB (3,3'-Diaminobenzidine)/Tris-HCl (pH7.6) 0.0003% Hydrogen peroxide in the dark. After the reaction was stopped, the embryos were washed in PBS, and placed in 4% PFA at 4 °C for long term storage.

### Imaging of chick embryos

For live-imaging, embryos were cultured in a 35 mm dish by the New culture method at 37 °C. Time-lapse images were recorded with Nikon TIRF/Spinning Disk microscope (Nikon Ti inverted fluorescent Microscope with CSU-22 spinning disc confocal) supported by Prime 95B Scientific CMOS camera (Photometrix), Nikon Widefield Epifluorescence inverted microscope (Nikon Ti inverted fluorescent Microscope with CSU-W1 large field of view) supported by ANDOR iXon camera and Nikon Eclipse TE2000-E supported by Hamamatsu ORCA-Flash 2.8 camera. The acquisition time was every 3 min using Nikon Elements Advance Research software V4.00.07. All time-lapse images were recorded every 3 min.

Immunostained chick embryos were imaged by Leica TCS SPE confocal microscope and Crest LFOV Spinning Disk with Nikon Ti2-E. Fixed ISH samples were imaged by Leica MZ16F microscope with Leica DFC300 Fx camera and Leica FireCam V.3.4.1 software.

## Quantification and statistical Analysis

### Cell number, mitotic rate, BrdU-, and Tunel-assay

To measure frequency of BrdU-positive or Tunel-positive cells, 3D-surface was created to an immunostained image of the embryonic disc by using the Surface tool in Imaris X64 9.2.0 software, and then the center of the embryonic disc was set for the following measurement. 3x3 squares with 800 μm of each side length were then arranged on the embryonic disc. Numbers of DAPI and BrdU-positive nuclei within each square were counted by Spots tool and averaged to get a frequency of BrdU- or Tunel-positive nuclei (*Figure 2F*, *Figure 1—figure supplement 1A*, and *Figure 2—figure supplement 2C*).

### Cell flow analyses

Cell flow analysis was performed with Imaris X64 9.2.0 software, Flowtrace and Particle Image Velocimetry (PIV) analysis. Frowtrace stacks every 20–40 frames from original live-imaging data and generates projection images of particle trajectories for visualizing cell movements. PIV analysis is described in the following part.

### PIV analysis and cell flow visualizing techniques

For quantitatively analyzing the cell flow pattern, the recorded images were further processed by using Particle Image Velocimetry (PIV) and subsequent visualization techniques. PIV analysis technique [PIVlab package in MATLAB (*Thielicke and Stamhuis, 2014*)], which is a common technique for mapping displacement of particles (e.g. tagged cells) over a short time interval measured from a series of consecutive images to obtain the velocity vector field is employed to quantify the cell flows

in the experimental time-lapse datasets. PIV is a standard technique used in the Fluid Physics and Engineering community to quantitatively measure and characterize fluid flows in different systems. This analysis provides comprehensive and quantitative information of a whole flow as a velocity vector field. The PIV analysis involves several steps that are carried out on image sequences over time intervals of 3 min, which is also the imaging acquisition speed in the experiments. First, we apply a high-pass filter during the image preprocessing step to highlight the bright particles (fluorescently tagged cell) and minimize the background. Then the PIV analysis was carried out by partitioning the image into smaller interrogation windows, typically of size 64x64 pixels with 50% overlap on the first pass, and a window size of 32x32 pixels with 50% overlap on the second pass. After this, raw velocity vector fields over time were obtained, and then these vectors were post-processed and validated by setting threshold limits, data smoothing and interpolation. A calibration was applied to convert pixel/frame units into μm/s units.

The generated velocity vector field through PIV were provided in *Figure 1—source data 1*, *Figure 2—source data 1*, *Figure 3—source data 1* and *Figure 4—source data 1*. These plots displayed the averaged cell pattern of the whole-flow-field in the certain time period in this study. To highlight the more specific rotating cell flows, mathematical/physical information (such as topology and vorticity) was extracted from these velocity vector fields via computational visualization techniques (*Thielicke and Stamhuis, 2014*; *Batchelor, 2000*; *Aris, 2012*; *Greenshields and Christopher, 2022*). Topology of the cell flow was determined by streamlines, which are a group of curves that approximate to contours of the vectors in the velocity vector field at a particular moment in time (*Batchelor, 2000*; *Aris, 2012*; *Greenshields and Christopher, 2022*). Flowtrace visualized the trajectories of the cell flow, whereas the streamlines illustrated the averaged flow pattern in the time period. Therefore, the trajectories and streamlines show similar pattern when the cell flow is in stable-state; however, they are not completely consistent with each other when the cell flow is unstable and time-dependently changing (*Batchelor, 2000*; *Aris, 2012*; *Greenshields and Christopher, 2022*). Vorticity, also termed as 'curl of vorticity', is a measure of a local rotation in the larger flow and its magnitude is twice that of the local angular velocity (*Batchelor, 2000*; *Aris, 2012*; *Greenshields and Christopher, 2022*). The vorticity plot further displays directionality of the rotation. Since vorticity arises from any rotational movements, including both a full- and partial rotation, the vorticity plot displayed the vorticities in both the vortex-like rotating and curves of the cell flow (*Thielicke and Stamhuis, 2014*; *Batchelor, 2000*; *Aris, 2012*; *Greenshields and Christopher, 2022*).

## Measurement of the distance between the left-right rotations of the polonaise movements in control-GFP and ΔDEP-embryos

The vector fields in control- and ΔDEP-GFP-embryos were generated from the recorded videos by using PIVlab and averaged the flow pattern of the polonaise movements for 6 hr. The centers of the left-right rotating cell flow of the polonaise movements in the averaged vector field were identified by streamlines and measured the distance between them by using ImageJ.

### Statistics

Statistical analyses were performed using two-tailed Student's t tests, one-way ANOVA in R (Ver3.5.0) and Excel 2022 (Microsoft). Statistical tests, individual p values, and sample numbers are described in figure panels and legends.

## Acknowledgements

We thank past and present Mikawa lab and Prakash lab members, particularly Drs J Hyer and L Hua for their invaluable suggestions and/or assistance. Imaging data for this study were acquired at the Center for Advanced Light Microscopy-CVRI at UCSF on microscopes obtained using funding from the Research Evaluation and Allocation Committee, the Gross Fund, and the Heart Anonymous Fund. This work was supported in part by grants from NIH (R01HL122375, R37HL078921, R01HL132832, R01HL148125) to TM; Uehara Memorial Foundation Fellowship and JSPS Postdoctoral Fellowship for Research Abroad to RA; Univ Miami funds to NPV; NSF CCC (DBI-1548297), CZI BioHub Investigator Program, and Howard Hughes Medical Institute to MP.

# Additional information

## Funding

| Funder | Grant reference number | Author |
| --- | --- | --- |
| National Heart, Lung, and Blood Institute | R01HL122375 | Takashi Mikawa |
| National Heart, Lung, and Blood Institute | R37HL078921 | Takashi Mikawa |
| National Heart, Lung, and Blood Institute | R01HL132832 | Takashi Mikawa |
| National Heart, Lung, and Blood Institute | R01HL148125 | Takashi Mikawa |
| Uehara Memorial Foundation | Fellowship | Rieko Asai |
| JSPS | Postdoctoral Fellowship for Research Abroad | Rieko Asai |
| University of Miami | start up funds | Vivek N Prakash |
| National Science Foundation | DBI-1548297 | Manu Prakash |
| Howard Hughes Medical Institute | | Manu Prakash |

The funders had no role in study design, data collection and interpretation, or the decision to submit the work for publication.

## Author contributions

Rieko Asai, Conceptualization, Data curation, Formal analysis, Validation, Investigation, Methodology, Writing – original draft, Writing – review and editing; Vivek N Prakash, Conceptualization, Formal analysis, Validation, Investigation, Methodology, Writing – original draft; Shubham Sinha, Formal analysis; Manu Prakash, Conceptualization, Supervision, Funding acquisition; Takashi Mikawa, Conceptualization, Supervision, Funding acquisition, Writing – original draft, Writing – review and editing

## Author ORCIDs

Rieko Asai ⓘ http://orcid.org/0000-0002-3826-8083
Vivek N Prakash ⓘ https://orcid.org/0000-0003-4569-6462
Shubham Sinha ⓘ http://orcid.org/0000-0003-4371-3070
Manu Prakash ⓘ http://orcid.org/0000-0002-8046-8388
Takashi Mikawa ⓘ http://orcid.org/0000-0003-1051-8260

Reviewer #1 (Public Review): https://doi.org/10.7554/eLife.89948.3.sa1
Reviewer #2 (Public Review): https://doi.org/10.7554/eLife.89948.3.sa2
Author response https://doi.org/10.7554/eLife.89948.3.sa3

# Additional files

## Supplementary files
• MDAR checklist

## Data availability

All data generated or analyzed during this study are included and provided in the manuscript, supplemental figures, Source data files, and videos. Raw datasets of immunofluorescent staining are available at Dryad at https://doi.org/10.5061/dryad.37pvmcvsx. Software used for PIV analysis is 'PIVlab', which is developed by William Thielicke and an open resource https://mathworks.com/matlabcentral/fileexchange/27659-pivlab-particle-image-velocimetry-piv-tool-with-gui; https://pivlab.blogspot.com.

The following dataset was generated:

| Author(s) | Year | Dataset title | Dataset URL | Database and Identifier |
|---|---|---|---|---|
| Asai R, Prakash Vivek N, Sinha S, Prakash M, Mikawa T | 2024 | Coupling and uncoupling of midline morphogenesis and cell flow in amniote gastrulation [Dataset] | https://doi.org/10.5061/dryad.37pvmcvsx | Dryad Digital Repository, 10.5061/dryad.37pvmcvsx |

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
