## [Editor Report · eLife assessment]

Large scale cell movements occur during gastrulation in vertebrate embryos but their role in this major morphogenetic transition in formation of the body plan is poorly understood. Using the chick embryo model system, this study makes **important** advances using elegant methods to show that extension of the primitive streak during gastrulation, occurring through cell proliferation, polarisation and intercalation, and large-scale polonaise cell movements, can be uncoupled. Although the driving mechanism and precise role of these movements remains a mystery, the study provides **convincing** evidence for the uncoupling through independent approaches, the most creative of which are the effects shown after induction of a supernumerary primitive streak.

---

## [Referee Report · Reviewer #1 (Public Review)]

In chicken embryos, the counter-rotating migration of epiblast cells on both sides of the forming primitive streak (PS), a process referred to as polonaise movements, has attracted longstanding interest as a paradigm of morphogenetic cell movements. However, the association between these cell movements and PS development is still controversial. This study investigated PS development and polonaise movements separately at their initial stage, showing that both could be uncoupled (at least at the initial phase), being activated via Vg1 signaling.

Strengths of this study

Polonaise movements, i.e., the circular cell migration of epiblast cells on both sides of the forming PS in avian embryos, have been the subject of research through live imaging and promoted the development of new tools to analyze quantitatively such movements. However, conclusions from previous studies remain controversial, at least partly due to the nature of perturbations to PS development and polonaise movements.

This study performed the challenging technique of electroporation to successfully mark and manipulate Wnt/PCP pathways in unincubated chicken embryo cells at the initiation phase of these two processes. In addition, the authors separately altered PS development and polonaise movements: PS development was perturbed by inhibiting either the Wnt/PCP pathway or DNA synthesis using aphidicolin, while polonaise movements were modified by the development of a second PS after engrafting Vg1-expressing COS cells located at the opposite end of the blastoderm. The study concluded that Vg1 elicits both PS development and polonaise movements, which occur in a parallel and are not inter-dependent.

To support these conclusions, particle image velocimetry (PIV) of cell trajectories captured by live imaging was performed. These tools delineated visually appealing cell movements and gave rise to vorticity profiles, adding more value to this study.

Weaknesses of this study

Engrafted Vg1-expressing COS cells located at the anterior end of the blastoderm elicited both the development of a second PS and marked bilateral polonaise movements while perturbing these movements along the original PS. How do polonaise movements along the second PS dominate over those along the normal PS? The authors suggested a model in which Vg1 acts in a graded or dose-dependent manner since engrafted COS cells over-expressed Vg1. This model can be tested by reducing the mass of engrafted COS cells. Although the authors propose performing this analysis in further investigations, it would be preferable to incorporate into this study for better consistency.

Thank you for indicating that this will be a focus of future studies.

---

## [Referee Report · Reviewer #2 (Public Review)]

Summary:

The authors are interested in large-scale cell flow during gastrulation and in particular in the polonaise movement. This movement corresponds to a bilateral vortex-like counter-rotating cell flow and transport the mesendodermal cells allowing ingression of cells through the primitive streak and ultimately the formation of the mesoderm and endoderm. The authors specifically wanted to investigate the coupling of the polonaise movement and primitive streak to understand whether the polonaise movement is a consequence of the formation of the primitive streak or the other way around. They propose a model where the primitive streak elongation is not required for the cell flow but rather for its maintenance and that robust cell flow is not required for primitive streak extension.

Strengths:

Overall, the manuscript is well written with clear experimental designs. The authors have used live imaging and cell flow analysis in different conditions, where either the formation of the primitive streak or the cell flow was perturbed.

Their live imaging and PIV-based analyses convincingly support their conclusions that primitive streak deformation or mitotic arrest do not impact the initiation of the polonaise movement but rather the location or maintenance of these rotations. They additionally showed that disruption of the polonaise movement in the authentic primitive streak by elegant addition of an ectopic primitive streak does not impact the original primitive streak elongation.

Weaknesses:

- Since myosin cables have been shown to be instrumental for the polonaise movement, it would be interesting to better investigate how the manipulations by the delta-DEP-GFP construct, or Vg1/Cos affect the myosin cables (as shown in preliminary form for the aphidicolin-treated embryos).

Thank you for indicating that this will be a focus of future studies.

---

## [Author Response]

The following is the authors’ response to the original reviews.

**Public Reviews:**

**Reviewer #1 (Public Review):**
In chicken embryos, the counter-rotating migration of epiblast cells on both sides of the forming primitive streak (PS), a process referred to as polonaise movements, has attracted longstanding interest as a paradigm of morphogenetic cell movements. However, the association between these cell movements and PS development is still controversial. This study investigated PS development and polonaise movements separately at their initial stage, showing that both could be uncoupled (at least at the initial phase), being activated via Vg1 signaling.Strengths of this studyPolonaise movements, i.e., the circular cell migration of epiblast cells on both sides of the forming PS in avian embryos, have been the subject of research through live imaging and promoted the development of new tools to analyze quantitatively such movements. However, conclusions from previous studies remain controversial, at least partly due to the nature of perturbations to PS development and polonaise movements.This study performed the challenging technique of electroporation to successfully mark and manipulate Wnt/PCP pathways in unincubated chicken embryo cells at the initiation phase of these two processes. In addition, the authors separately altered PS development and polonaise movements: PS development was perturbed by inhibiting either the Wnt/PCP pathway or DNA synthesis using aphidicolin, while polonaise movements were modified by the development of a second PS after engrafting Vg1-expressing COS cells located at the opposite end of the blastoderm. The study concluded that Vg1 elicits both PS development and polonaise movements, which occur in a parallel and are not inter-dependent.To support these conclusions, particle image velocimetry (PIV) of cell trajectories captured by live imaging was performed. These tools delineated visually appealing cell movements and gave rise to vorticity profiles, adding more value to this study.Weaknesses of this studyEngrafted Vg1-expressing COS cells located at the anterior end of the blastoderm elicited both the development of a second PS and marked bilateral polonaise movements while perturbing these movements along the original PS. How do polonaise movements along the second PS dominate over those along the normal PS? The authors suggested a model in which Vg1 acts in a graded or dose-dependent manner since engrafted COS cells over-expressed Vg1. This model can be tested by reducing the mass of engrafted COS cells. Although the authors propose performing this analysis in further investigations, it would be preferable to incorporate into this study for better consistency.

We would like to express our gratitude to the editors and the reviewers for finding the valuable significances of our study and for giving thoughtful suggestions. We agree that it would be a logical next step to identify the driving mechanism(s) of the polonaise movements, although this is beyond the scope of the current study. Rather, it is the focus of ongoing studies, in which we are investigating how Vg1 works in this concentration context and resulting dose-dependent effect on downstream gene expression, in order to provide a comprehensive understanding of this interesting dual role of Vg1. The relationship between the intensity of Vg1 signaling and the polonaise movements can be tested by modifying the size of the Vg1/COS, as the reviewer pointed out.

The authors claim that chicken embryo development is representative of "amniotes," but it does not hold for all groups. Avian and mammal species are exceptional among amniotes in the sense they develop a PS (e.g., Coolen et al. 2008). Moreover, in certain mammalian embryos like mouse embryos, cells laterally to the PS do not move much (Williams et al. 2012). The authors should avoid the generalization that chicken embryos unequivocally represent amniotes as opposed to the observed in non-amniote embryos. The observations in chicken embryos as they stand are significant enough.References:Coolen M, et al. (2008). Molecular characterization of the gastrula in the turtle Emys orbicularis: an evolutionary perspective on gastrulation. PLoS One. 3(7):e2676. doi: 10.1371/journal.pone.0002676Williams M, et al. (2012). Mouse primitive streak forms in situ by initiation of epithelial to mesenchymal transition without migration of a cell population. Dev Dyn. 241(2):270-283. doi: 10.1002/dvdy.23711

We modified the following sentences to the summary and introduction of the revised version as below:

In Summary:

(p.1, Lines 9-11.) “Large-scale cell flow characterizes gastrulation in animal development. In amniote gastrulation, particularly in avian gastrula, a bilateral vortex-like counter-rotating cell flow, called ‘polonaise movements’, appears along the midline.”

In Introduction:

(p.2, Lines 43-46.) “In amniotes, particularly in avian gastrula (i.e. embryonic disc), a bilateral vortex-like counter-rotating cell flow, termed ‘polonaise movements’, occurs within the epiblast along the midline axis, prior to and during primitive streak (PS) formation.”

**Reviewer #2 (Public Review):**
Summary:The authors are interested in large-scale cell flow during gastrulation and in particular in the polonaise movement. This movement corresponds to a bilateral vortex-like counter-rotating cell flow and transport the mesendodermal cells allowing ingression of cells through the primitive streak and ultimately the formation of the mesoderm and endoderm. The authors specifically wanted to investigate the coupling of the polonaise movement and primitive streak to understand whether the polonaise movement is a consequence of the formation of the primitive streak or the other way around. They propose a model where the primitive streak elongation is not required for the cell flow but rather for its maintenance and that robust cell flow is not required for primitive streak extension.Strengths:Overall, the manuscript is well written with clear experimental designs. The authors have used live imaging and cell flow analysis in different conditions, where either the formation of the primitive streak or the cell flow was perturbed.Their live imaging and PIV-based analyses convincingly support their conclusions that primitive streak deformation or mitotic arrest do not impact the initiation of the polonaise movement but rather the location or maintenance of these rotations. They additionally showed that disruption of the polonaise movement in the authentic primitive streak by elegant addition of an ectopic primitive streak does not impact the original primitive streak elongation.Weaknesses:When using the delta-DEP-GFP construct, the authors showed that they can manipulate the shape of the primitive streak without affecting the identity and number of primitive streak cells. It is not clear however how this can affect the shape, volume or adhesion of the cells. Some mechanistic insights would strengthen the paper.

We appreciate the reviewer’s invaluable feedback. We agree that it would be informative to know how the ΔDEP-GFP construct led to PS deformation. This approach has been previously introduced by Voiculescu et al., (2007) to demonstrate an involvement of the Dsh(DEP) in PS shape regulation as described in text (please see pp4-5, lines 91-94 in Results and p13, lines 279-281 in Discussion). The previous study suggested that the Wnt/PCP pathway through Dsh(DEP) is a major regulator of cell intercalation, which plays an important role in PS morphogenesis (Voiculescu et al., 2007).

Overall, frequencies of observation are missing for a better view of the phenomenon. For example, do Vg1/Cos cells always disrupt the flow at the authentic primitive streak? Can replicate vector fields be integrated to reflect quantification?

We agree and have added the numbers of embryos examined. In our experimental system, the Vg1/COS-implanted embryos always exhibited that the original polonaise movements along the authentic PS were always disrupted by the induced polonaise movements (n=4/4 embryos). The replicated vector fields were integrated to the Streamline and Vorticity plots (please see Fig. 1-4, Fig. S1, S4-7).

Since myosin cables have been shown to be instrumental for the polonaise movement, it would be interesting to better investigate how the manipulations by the delta-DEP-GFP construct, or Vg1/Cos affect the myosin cables (as shown in preliminary form for the aphidicolin-treated embryos).

We agree that investigations of cytoskeletons and motor proteins would provide deeper understandings as to how the ΔDEP-GFP construct and perhaps Wnt/PCP components work in PS formation and morphogenesis. We plan to examine, as a future study, the patterns of the myosin cables in the ΔDEP-GFP-misexpressing or Vg1/COS-implanted embryos to get better understanding the mechanism(s) of the polonaise movements as the reviewer pointed out.

**Recommendations for the authors:**

**Reviewer #1 (Recommendations For The Authors):**
The authors named the dominant-negative Dsh lacking DEP [dnDsh(deltaDEP)]-fused GFP as deltaDEP-GFP, presumably to distinguish it from the construct dnDsh-deltaPDZ previously reported. However, the prefix "dnDsh" conveys the critical function in the present study. The reviewer recommends spelling out dnDsh(deltaDEP)-GFP to clarify to readers which signal was manipulated.

We agree that it is necessary to distinguish our construct used in this study from the dnDsh-deltaPDZ construct. We have, therefore, clarified the abbreviation in the main text as follows (please see pp 4-5, lines 91-97): ‘The DEP domain of Dishevelled (Dsh; a transducer protein of Wnt signaling) is responsible for the non-canonical Wnt/PCP pathway (43, 44), and misexpression of dominant-negative Dsh lacking DEP [dnDsh(ΔDEP)] leads to deformation of the midline structures, including the PS (21). Further, the Wnt/PCP pathway is involved in cellular polarity and migration, while the canonical Wnt pathway regulates cell proliferation (45). We refer the dnDsh(ΔDEP)-GFP construct that we generated, as ΔDEP-GFP, and tested its ability to alter cellular polarity, resulting in PS deformation’.

The authors described the "Vg1 plasmid DNA" as a gift from Claudio D. Stern and Jane Dodd. However, they should indicate the vector backbone, especially whether the vector carries the SV40 ori sequence. Ori-containing plasmids multiply after transfection as COS cells express the SV40T antigen, leading to protein overexpression.

We added the name of the plasmid ‘pMT23-Vg1-myc-GDF1’ to the ‘Material and methods’ section (please see p25, line 574). pMT23 expression vector is a derivative of pMT21 (Hume and Dodd, 1993) and contains SV40 ori (Wong et al., 1985).

**Reviewer #2 (Recommendations For The Authors):**
Most of the comments are indicated in the public review.There are additionally minor modifications that would help readers interpret the figures. In Figure S1B and D, it is not clear to the reader what the asterisks indicate.

We added the sentence ‘The white asterisks indicate GFP-expressing cells.’ to the figure legend of the Fig. S1 B and D (please see p34, line 874).